# Payload Delivery: Engineering Immune Cells to Disrupt the Tumour Microenvironment

**DOI:** 10.3390/cancers13236000

**Published:** 2021-11-29

**Authors:** Daniel Fowler, Callum Nattress, Alba Southern Navarrete, Marta Barisa, Jonathan Fisher

**Affiliations:** UCL Great Ormond Street Institute of Child Health, 20 Guilford St, London WC1N 1DZ, UK; d.fowler@ucl.ac.uk (D.F.); callum.nattress.19@ucl.ac.uk (C.N.); a.southern@ucl.ac.uk (A.S.N.); m.barisa@ucl.ac.uk (M.B.)

**Keywords:** cancer, solid tumours, cell therapy, immunotherapy, engineered immune cells, CAR T cell, armoured CAR T cell, payload delivery, tumour microenvironment, immunosuppression

## Abstract

**Simple Summary:**

Cell-based therapies composed of genetically engineered immune cells have huge potential for treating cancer. T cells modified to express tumour-targeting receptors have been proven to be highly effective against blood cancers but not solid malignancies. Using innovative genetic modifications, we have the potential to enhance solid tumour targeting by cell-based therapies and fulfil this unmet clinical need. Different genetic engineering approaches have been proposed, each designed to overcome specific hurdles associated with solid tumours. In this review, we discuss the novel ways of engineering immune cells to enhance efficacy against solid tumours, and how this has led to a new era of cell-based therapies, capable of delivering immunotherapeutic payloads that disrupt the immunosuppressive tumour microenvironment and generate concerted anti-tumour immune responses.

**Abstract:**

Although chimeric antigen receptor (CAR) T cells have shown impressive clinical success against haematological malignancies such as B cell lymphoma and acute lymphoblastic leukaemia, their efficacy against non-haematological solid malignancies has been largely disappointing. Solid tumours pose many additional challenges for CAR T cells that have severely blunted their potency, including homing to the sites of disease, survival and persistence within the adverse conditions of the tumour microenvironment, and above all, the highly immunosuppressive nature of the tumour milieu. Gene engineering approaches for generating immune cells capable of overcoming these hurdles remain an unmet therapeutic need and ongoing area of research. Recent advances have involved gene constructs for membrane-bound and/or secretable proteins that provide added effector cell function over and above the benefits of classical CAR-mediated cytotoxicity, rendering immune cells not only as direct cytotoxic effectors against tumours, but also as vessels for payload delivery capable of both modulating the tumour microenvironment and orchestrating innate and adaptive anti-tumour immunity. We discuss here the novel concept of engineered immune cells as vessels for payload delivery into the tumour microenvironment, how these cells are better adapted to overcome the challenges faced in a solid tumour, and importantly, the novel gene engineering approaches required to deliver these more complex polycistronic gene constructs.

## 1. Introduction

Harnessing the inherent anti-tumour properties of our immune system is a promising therapeutic strategy for cancer, not only as a last resort for patients that are refractory to standard treatments, but also as a stand-alone therapy in and of itself. A particularly remarkable attribute of our immune system is that certain immune cells can detect malignantly transformed cells and use sophisticated cytotoxic machinery to deliver a death signal that is both potent and targeted [1,2]. In addition, they are capable of navigating the labyrinth of blood vessels that run throughout the human body, extravasating towards local inflammatory cues associated with tumours and squeezing through tight interstitial spaces in order to penetrate deep inside cancerous tissues [3]. By exploiting these immune cells, we have the exciting potential to create “living drugs”, sentinel-like cell-based therapies that patrol a patient’s body, seeking out malignant cells and eradicating them whilst leaving healthy tissues unharmed.

Despite immune cells offering a certain degree of natural protection against cancer, tumours have evolved counter defence mechanisms that enable them to evade and/or suppress anti-tumour effector responses [4]. This allows tumours to overcome the host’s inherent immune defences, and more importantly, cell-based immunotherapies that use non-engineered immune cells. Indeed, with some notable exceptions such as the anti-melanoma efficacy of ex-vivo expanded tumour-infiltrating lymphocytes (TILs) [5], and the graft-versus leukaemia phenomenon observed in allogenic haematopoietic stem cell transplantation (HSCT) [6], clinical trials have tested the efficacy of various non-engineered cell-based products and have yet to demonstrate robust clinical responses. In light of these findings, it has become widely accepted that non-engineered immune cells are unable to overcome tumour-associated immunosuppression, and efforts are currently focussed on a new and exciting era of genetically modified immune cells for adoptive cell therapy.

With the advent of modern gene engineering techniques, it is now possible to genetically alter immune cells for adoptive cell therapy. Gene modification can be used to insert completely new genes into the recipient cell’s genome, allowing the cell to produce proteins it otherwise would not express, or repeated inserts of an existing gene, now under the control of a potent promoter, which results in enhanced and/or constitutive expression. This has the potential to bolster immune cell effector responses against evasive/suppressive tumours and to improve the overall efficacy of cell-based therapies against cancer. Genes encoding chemokine receptors that facilitate tumour homing [7,8,9], for example, or receptors that confer potent cytotoxicity against tumour cells [10], or cytokines that promote survival and orchestrate bystander cells [11], are just some of the potential ways in which anti-tumour immunity can be enhanced. The possibilities and scope are seemingly limitless.

The therapeutic potential of engineered immune cells has been well demonstrated using polyclonally expanded T cells genetically modified to express chimeric antigen receptors (CARs) composed of tumour-targeting ectodomains coupled with powerful stimulatory endodomains [12]. The resulting T cells have a re-directed specificity and reactivity that renders them highly cytotoxic against cells expressing the target antigen. CAR T cells targeting the B cell lineage marker CD19 have proven highly successful in generating long-term remission in B cell lymphoma and acute lymphoblastic leukaemia [13], and have the approval of both the Food and Drug Administration (FDA) and the Medicines and Healthcare products Regulatory Agency (MHRA) for use in these indications. Kite’s CD19-targeting CAR T cell product Tecartus^TM^, for example, induced impressive durable remissions in phase II clinical trials and has recently been approved by the FDA for use in relapsed or refractory mantle cell lymphoma [14].

Despite CAR T cells being highly efficacious against certain haematological malignancies, optimisation is still ongoing in areas such as antigen selection, outgrowth of antigen negative tumours, exhaustion, on-target off-tumour cytotoxicity, graft rejection, and cytokine release syndrome (all covered elsewhere in more focussed reviews on CAR T cell efficacy [15,16,17]). Most importantly, the demonstrable success of CAR T cell therapies against haematological malignancies has yet to be recapitulated in non-haematological “solid” tumours, which are more prevalent and notoriously difficult to treat. Compared with haematological malignancies, solid tumours pose many additional challenges that blunt CAR T cell potency, including homing, immunosuppression, nutrient deprivation, low pH, and hypoxia [18]. Cell-based therapies capable of overcoming these hurdles are an unmet clinical need and the search for additional more sophisticated engineering strategies is ongoing.

In this review, we discuss the challenges posed by a solid tumour, how they can hamper cell-based immunotherapies, and the novel gene modifications for improving efficacy. Gene constructs encoding a variety of different surface bound and/or secreted proteins are discussed, which confer more than just redirected cytotoxicity and redefine engineered immune cells as vessels for immunotherapeutic payload delivery, capable of orchestrating potent anti-tumour immunity at range. In addition, we discuss key engineering techniques required to deliver the polycistronic gene cassettes required to create this next generation of cell-based therapies.

## 2. Solid Tumour Targeting by Engineered Immune Cells

### 2.1. Solid Tumour Homing

In order for CAR T cells to deliver their death signal to tumour cells, they must engage directly with the tumour cell membrane and form a stable cytotoxic immunological synapse. This requires colocalization between CAR T cells and tumour cells, which is governed by three key factors: (a) route of administration, (b) anatomical location of the tumour, and most importantly, (c) CAR T cell homing in vivo.

CAR T cells are typically administered to patients via intravenous (IV) infusion. In the context of haematological malignancies, where the bulk of the tumour burden resides in the peripheral blood, IV infusion delivers effector cells directly to the site of the tumour, and thus requires minimal homing. In contrast, solid tumours are tissue-specific, occupying discrete anatomical locations towards which CAR T cells must actively travel. This additional challenge may explain—at least in part—why CAR T cells are highly efficacious against blood-borne cancers, yet are poorly effective at treating solid malignancies [19].

Solid tumours are rapidly dividing masses of cells with an ever-increasing demand for oxygen and nutrients [20]. By secreting pro-angiogenic factors, such as vascular endothelial growth factor (VEGF) and angiopoietin, which recruit endothelial cells and promote vascularisation, tumours are able to wire themselves directly into the host’s vascular network and subsequently tap into its nutrient supply [21,22]. Vascularisation of a solid tumour provides the all-important access required for an intravenously administered cell-based therapy, provided it is programmed with the correct tissue homing required to exit the circulation at the precise anatomical location of the tumour.

Tissue homing requires a multistep process consisting of endothelial tethering, extravasation, and interstitial migration. Each step involves synchronised interactions between chemotactic molecules and their cognate receptors, including addressins, integrins, adhesion molecules, cytokines, and chemokines (reviewed elsewhere [23]). Chemokines are particularly important for tissue homing by circulating immune cells [24], and a number of chemokines and their cognate receptors have been implicated in homing towards solid tumours [25]. The inflammatory chemokine receptors CCR2, CXCR3, and CCR5, for example, are associated with T cell infiltration in lung adenocarcinoma, colorectal carcinoma, and metastatic melanoma [26,27,28]. Moreover, their cognate ligands, including CCL2, CCL3, CCL4, CCL5, CXCL9, CXCL10, and CXCL11, are often upregulated in a range of different tumour types [29,30,31,32,33]. 

Engineering CAR T cells to stably express selected chemokine receptors has the potential to greatly enhance their tumour homing in vivo and subsequently improve their overall efficacy against solid malignancies. Preclinical studies have already tested the co-expression of selected tumour-homing genes in CAR T cells. For example, the surface integrin αvβ6 and myeloid-derived chemokine IL-8 are frequently over expressed in many types of solid tumours, and the dual expression of IL-8 receptors—either CXCR1 or CXCR2—in αvβ6-targeting CAR T cells has been proposed as a potential strategy for enhanced solid tumour homing [7]. In addition, solid tumours often overexpress mesothelin on their surface and produce high levels of macrophage chemotactic protein (MCP)-1, providing a strong rationale for co-expressing the MCP-1 receptors CCR2b or CCR4 in mesothelin-targeting CAR T cells, as tested by Wang et al. [8]. Furthermore, Liu et al. focussed specifically on hepatocellular carcinoma and reported that glypican-3 and CXCR2 ligands are commonly overexpressed in this tumour type. Accordingly, the authors co-expressed CXCR2 in glypican-3-targeting CAR T cells for enhanced hepatocellular carcinoma homing [9]. In all of these studies, the co-expression of a chemokine receptor was feasible with no adverse effect on CAR expression or function. Importantly, cells expressing both CAR and the chemokine receptor showed enhanced migration towards their cognate ligands in vitro, as well as better homing and efficacy against solid tumours in in vivo xenograft mouse models.

Despite promising preclinical data, we have yet to see whether or not this strategy translates into improved efficacy against solid malignancies in clinical trials. No doubt tumour infiltration by circulating immune cells will differ somewhat in murine models compared with in situ. Furthermore, tumours can often be avascular or contain vasculature that is abnormal, poorly organised, and leaky, which could significantly impede immune cell infiltration regardless of chemokine receptor modification. Indeed, normalisation of the haphazardly organised tumour vasculature using anti-VEGF has been shown to improve immune cell infiltration [34]. Added to this, there is the further complexity of tumours using epigenetic silencing to switch off production of Th1-based chemokines such as CXCL9 and CXCL10, a novel immune escape mechanism employed by tumours to reduce infiltration by cytotoxic effector cells [35]. Nonetheless, data demonstrate the feasibility of a dual expression of genes for cytotoxicity and homing, paving the way for a new generation of CAR T cell therapies that are tailored for improved tumour infiltration, and thus better efficacy against solid cancers (Figure 1).

### 2.2. Overcoming Immunosuppression

Solid tumours are not only challenging for CAR T cells to infiltrate, they are also a highly immunosuppressive arena in which cell-mediated cytotoxicity must take place. Metabolic obstacles including hypoxia, low pH, and depletion of essential nutrients such as amino acids and glucose, are common features of the solid tumour microenvironment (TME) that can markedly suppress immune cell function [36,37,38]. Further obstacles are posed by immunosuppressive cell types such as tumour-associated macrophages (TAMs), T regulatory cells (Tregs), and myeloid-derived suppressor cells (MDCSs) [39,40]; these cells express high levels of immune inhibitory molecules, such as programmed death receptor ligands (PD-L1 and PD-L2), galectin-9, and the so-called herpes virus entry mediator (HVEM), and secrete large amounts of anti-inflammatory cytokines, including transforming growth factor-β (TGF-β and interleukin (IL)-10 [39,41,42,43,44,45,46,47].

Collectively, the TME is highly proficient at switching off anti-tumour immunity, enabling solid tumours to grow and invade adjacent tissues unchecked. This immunosuppressive barrier is another contributing factor to the limited efficacy of CAR T cells against non-haematological cancers [48], and the co-expression of additional genes to help CAR T cells overcome specific immunosuppressive components of the TME could substantially enhance their success against solid malignancies. Thus far, efforts have focused on the following gene products: (a) proteins that counteract metabolic immunosuppression, (b) CARs that are specific for antigens associated with non-malignant tumour-associated immunosuppressive cells, and (c) secreted proteins that neutralise inhibitory cytokines and/or cell surface proteins.

#### 2.2.1. Counteracting Metabolic Immunosuppression

The metabolism has emerged recently as having a profound influence over immune cell function. Immune cells that are adapted to survive and function within the adverse metabolic conditions of the TME are therefore exciting candidates for cell-based therapies against solid tumours. Although only in the early stages of development, prototypic engineered immune cells that co-express metabolic enhancers for improved solid tumour targeting have already been designed with promising preliminary results reported to date.

Hypoxia is a hallmark of solid tumours that contributes significantly to immunosuppression within the TME [49]. Disorganised cell growth, metabolism, and vascularisation contribute to reduced oxygen availability within tumours, ultimately leading to increased levels of reactive oxygen species—namely hydrogen peroxide (H_2_O_2_)—and a state of oxidative stress within the TME that thwarts anti-tumour immunity [36]. To counteract this, Ligtenberg et al. engineered CAR T cells with genes that encode catalases, enzymes which metabolise H_2_O_2_ into water and oxygen, and thus ameliorate oxidative stress. These CAR T cells displayed lower levels of oxidative stress following activation and, importantly, were able to sustain their cytotoxic function and boost that of bystander T cells and natural killer (NK) cells in the presence of high H_2_O_2_ [50].

The hypoxic conditions of the TME can be exploited as a tumour-associated trigger for engineered T cells. Kosti et al., for example, engineered CAR T cells with an oxygen-dependent degradation domain from the hypoxia-inducible factor 1-α (HIF1-α transcription factor, which acts as an oxygen-sensing safety switch [51]. The CAR is ubiquitinated and degraded under normoxic conditions, whereas under hypoxia, the CAR is expressed. In xenograft mouse models, HIF1-α co-expression in CAR T cells targeting the receptor tyrosine kinase ErbB, which is expressed in both tumour and healthy organs, resulted in selective expansion of ErbB-targeting CAR T cells in the tumour, and ultimately tumour rejection with no associated on-target off-tumour toxicity.

In addition, engineering approaches that bolster immune cell function in arginine low conditions have also been tested. Arginine availability is often very low in solid tumours due to increased consumption by rapidly dividing tumour cells and the overexpression of arginase-1 on tumour-infiltrating cells such as MDSCs [52]. Arginine is essential for optimal immune cell function and low arginine concentrations in the surrounding milieu can markedly dampen effector responses [38]. To offset this, Fultang et al. have proposed that co-expressing the arginine resynthesis enzymes argininosuccinate synthase and ornithine transcarbamylase in CAR T cells can support effector function in the arginine low conditions of the TME [53]. The authors report increased CAR T cell proliferation, sustained cytotoxicity, no concomitant exhaustion, and better efficacy against leukemic and solid tumour models.

#### 2.2.2. Targeting Tumour-Resident Immunosuppressive Cells

The abundant populations of TAMs, Tregs, and MDSCs often found in solid tumours are potent at suppressing immune cell function via multiple mechanisms of immunosuppression, including the following: (1) secretion of TGF-β, which inhibits effector cell proliferation, differentiation, and cytotoxicity [54]; (2) secretion of IL-10, an anti-inflammatory cytokine that downregulates major histocompatibility complex (MHC) II, blocks Th1 cytokine production, and inhibits NFκB signalling [55,56]; (3) expression of checkpoint ligands, which switch off activated effector cells [57]; (4) secretion of tissue remodelling matrix metalloproteinases (MMPs) that facilitate tumour metastasis and invasion [58]; and (5) secretion of chemokines such as CCL17 and CCL22, which recruit more immunosuppressive cells leading to the maintenance of the TME [59], as well as CXCL8, which further supports tumour metastasis and invasion [60]. Targeting immunosuppressive cells within the tumour, as well as the tumour cells themselves, has the potential to knock out all of the above mechanisms of immunosuppression and thus enhance CAR T cell efficacy against solid malignancies.

Prototypic CAR T cells that target unique antigens on the surface of non-malignant pro-tumourigenic immune cells prevalent in the TME are under development. These CARs can be either co-expressed with tumour-targeting CARs, or can be exclusively expressed and administered on their own or co-administered in conjunction with conventional tumour-targeting CAR T cells. Rodriguez-Garcia et al., for example, generated murine CAR T cells specific for folate receptor beta (FRβ), a receptor that is often highly expressed on human and murine TAMs [61]. In syngeneic mouse models, these cells selectively eliminated TAMs, resulting in a marked enrichment of pro-inflammatory macrophage subsets within the TME, enhanced CD8^+^ T cell infiltration, reduced tumour growth, and prolonged overall survival. Interestingly, pre-conditioning the TME using FRβ-targeting CAR T cells markedly enhanced the efficacy of mesothelin-targeting CAR T cells. Further research in this field will continue to unveil novel targetable markers associated with TAMs, as well as the closely related MDSCs.

CARs that target Tregs have also been explored. Although this has the potential to eliminate a significant immunosuppressive component of the TME, identifying an appropriate Treg-specific antigen is particularly challenging, especially considering that these cells share many of the same markers as not only the CAR T cells themselves, but also bystander effector T cells. Preliminary data from Dehbashi et al. demonstrated that CD25-targeting CAR NK cells could be generated using the NK cell line NK-92, and that these cells were cytotoxic towards CD25-expressing Jurkat cells in vitro [62]. CD25 is the high affinity IL-2Rα, which is constitutively expressed at high levels on Tregs, yet only moderately and transiently expressed on activated T cells and NK cells. Alternatively, Perera et al. identified CCR4 as being over-expressed on CD4^+^CD25^+^Foxp3^+^ Tregs that accumulate in tumours, and thus generated CCR4-targeting CAR T cells that were potently cytotoxic against CCR4-expressing cell lines [63]. However, whether or not these cells can selectively eliminate Tregs in the TME and whether this has any meaningful effect on tumour growth without concomitant toxicity has yet to be seen.

#### 2.2.3. Blocking Immune Checkpoints

Immune checkpoints are inherent safety features of the immune system that play a critical role in switching off redundant immune responses, thus preventing damage to healthy tissues after an infection has been cleared. Typically, immune checkpoints are receptors expressed on the surface of activated immune cells that deliver an inhibitory or apoptosis-inducing signal upon ligand binding; for example, PD-1 is often upregulated on activated T cells, and upon binding its cognate ligands PD-L1 and PD-L2, induces anergy and/or apoptosis in the cell on which it is expressed [57]. Others include cytotoxic T lymphocyte-associated antigen-4 (CTLA-4), lymphocyte activation gene-3 (LAG-3), T cell immunoglobulin and mucin-domain containing-3 (TIM-3), T cell immunoglobulin and ITIM domain (TIGIT), and V-domain Ig suppressor of T cell activation (VISTA) [64].

Immune checkpoints play a notable role in suppressing anti-tumour immunity and thus in facilitating tumour escape; indeed, the PD-1-targeting checkpoint inhibitors Pembrolizumab and Nivolumab have shown remarkable success in the treatment of non-small cell lung cancer (NSCLC), melanoma, bladder cancer, and Hodgkin’s lymphoma [65,66,67,68,69,70,71]. Moreover, combinatorial approaches with CTLA-4-targeting checkpoint inhibitors such as Ipilimumab have shown considerable advantages over monotherapy [72]. Tumour cells and tumour-associated myeloid cells such as TAMs and MDSCs can express high levels of checkpoint ligands such as PD-L1 and PD-L2 [73], which can potentially switch off tumour-infiltrating immune cells—particularly CAR T cells—before they get the chance to exert their cytotoxic effect.

To counter this, cell-based therapies could be combined with checkpoint inhibitors such as Pembrolizumab; indeed, preliminary results suggest a potential synergy in both preclinical experiments and early phase clinical trials [74,75]. A number of studies have also tested the feasibility of engineering CAR T cells to co-express secretable antibody or antibody-like molecules capable of neutralising PD-1 signalling. In contrast with combination therapy, this has the benefit of a targeted release, higher local concentrations, continuous production, and less interventions for the patient.

Some studies have focussed on secretable molecules that block PD-L1. Saurez et al., for example, demonstrated that carbonic anhydrase IX (CAIX)-targeting CAR T cells could be engineered to co-express genes encoding neutralising IgG1 or IgG4 antibodies against PD-L1 [76]. These CAR T cells constitutively secreted PD-L1 blocking antibody, expressed lower levels of markers associated with T cell exhaustion such as LAG-3, TIM-3, and PD-1 following antigen stimulation in vitro, and importantly, displayed enhanced efficacy in an orthotopic in vivo model of CAIX-positive hepatocellular carcinoma. In addition, Xie et al. generated EIIIB-targeting CAR T cells that secrete PD-L1-neutralising single domain antibody fragments [77]. These CAR T cells, which target an alternatively spliced domain of fibronectin highly expressed in tumours, displayed a reduced expression of PD-1, CTLA-4, TIM-3, and TIGIT, and improved persistence and expansion in syngeneic mouse models of melanoma. They did not, however, improve survival compared with CAR only, which could be attributed to the relatively low secretion rates of the checkpoint inhibitor.

Alternatively, secretable molecules that target PD-1 itself have been explored. Rafiq et al., for example, generated CD19-targeting CAR T cells that co-express secretable PD-1 neutralising scFv fragments [78]. Using syngeneic and xenograft models, the authors demonstrated an enhanced activity of CAR T cells, as well as that of bystander tumour-specific T cells. In studies more relevant to solid tumour targeting, co-expression of secretable PD-1 neutralising scFv in epidermal growth factor receptor (EGFR)- and mesothelin-targeting CAR T cells has been tested. These cells showed enhanced proliferation in vitro and improved overall efficacy in murine xenograft models [79,80]. Furthermore, in a case study reported by Fang et al., prolonged survival was observed in an advanced refractory ovarian cancer patient treated with mesothelin-targeting CAR T cells that secrete anti-PD-1 full length antibody [81].

#### 2.2.4. Neutralising Immunosuppressive Cytokines

Immunosuppressive cytokines are powerful dampeners of anti-tumour immune responses, which, like PD-1, are promising targets for CAR T cell armouring. Of particular interest is the Treg- and TAM-derived cytokine TGF-β, an abundant protein within the TME of solid tumours that has potent inhibitory effects on multiple T cell effector functions, including proliferation, differentiation, cytokine production, and cytotoxicity [82].

Gene constructs that encode TGF-β-neutralising molecules, such as trap proteins and decoy receptors, which bind and sequester TGF-β without downstream signalling, have shown early promise in preclinical studies. Li et al., for example, co-expressed a TGF-β trap protein consisting of a TGF-βR ectodomain in EGFRvIII-targeting CAR T cells, rendering them resistant to TGF-β and subsequently improving their survival and efficacy in preclinical models of glioblastoma [83]. Interestingly, glioblastoma-associated microglia in the tumours from treated mice expressed elevated levels of markers for a pro-inflammatory M1 macrophage phenotype, suggesting macrophage repolarisation within the tumour as a potential mechanism of action. Similarly, Chen et al. engineered CAR T cells that secrete bispecific trap proteins consisting of an anti-PD-1 scFv fused with a TGF-βRII ectodomain [84]. These armoured CAR T cells were able to simultaneously neutralise surface bound PD-1 and soluble TGF-β, resulting in improved effector function in vitro and enhanced efficacy in xenograft mouse models compared with the CAR only or CAR plus anti-PD-1-secretion controls. Furthermore, Kloss et al. engineered prostate-specific membrane antigen (PSMA)-targeting CAR T cells to co-express a dominant-negative TGF-βRII, a dud receptor that mops up TGF-β without intracellular signalling. These CAR T cells conferred enhanced survival and efficacy in an in vivo model of prostate cancer [85] and are currently undergoing a first in human clinical trial for refractory castration-resistant metastatic prostate cancer (NCT03089203).

Particularly interesting is an alternative approach proposed by Chang et al. [86]. This group designed CARs that target soluble ligands, specifically CARs with a TGF-β-specific scFv ectodomain coupled with a CD28ζ endodomain. T cells engineered with these CARs can potentially promote anti-tumour responses in two ways: (1) by sequestering TGF-β within the TME, thus preventing its inhibitory effects on tumour-infiltrating effector cells, and (2) by producing Th1 cytokines such as IFN-γ, TNF-α, and IL-2 in response to TGF-β, thus converting/switching this inhibitory cytokine into an activatory signal, thus counteracting immunosuppression within the TME. The feasibility of this approach has been explored thoroughly; however, we have yet to see how efficacious these cells are in vivo.

As demonstrated in these preclinical studies, engineering CAR T cells with the added ability to disrupt tumour immunosuppression has the potential to improve efficacy against solid tumours (Figure 1). It is important to acknowledge, however, that tumours can exploit many different suppressive mechanisms to evade immune responses, and identifying the rate limiting pathways of suppression within solid tumours is an ongoing challenge for cell-based therapies.

### 2.3. Recruiting Bystander Cells

CAR T cells that manage to infiltrate a solid tumour and then subsequently overcome the immunosuppressive TME still have the difficult challenge of eradicating a rapidly dividing mass of malignant cells. Due to the added complexity of immune cell homing, it is likely that only a small fraction of adoptively transferred T cells successfully infiltrate a solid tumour. Furthermore, CAR-expressing cells only represent a minor fraction of the total T cell pool injected into the patient, and given that the functional benefits of a CAR are strictly limited to the T cells that express it, the potential number of tumour-infiltrating effector cells is realistically very low. Therefore, in the context of solid malignancies, CAR T cells are likely to be vastly outnumbered by tumour cells and will need to self-amplify and serial kill in order to have any significant effect in the clinic.

A promising strategy to further support CAR T cells in their fight against solid tumours is to engineer them with the added ability to recruit bystander immune cells. With transduction efficiencies for CAR T cell therapies being as low as 30%, the vast majority of T cells infused into the patient are unable to recognise tumour cells, yet they express the cytotoxic machinery capable of exerting potent anti-tumour responses. These cells may be recruited by chemokines released from tumour-infiltrating CAR T cells, or indeed, may infiltrate the tumour of their own accord. In addition, there is huge anti-tumour potential within the patient’s own endogenous immune system, which includes CD8^+^ T cells, NK cells, γδ T cells, macrophages, and neutrophils. Although potentially recovering from lympho-depleting chemotherapy, it too can be mobilised against the tumour provided it is given the right support.

The non-transduced component of CAR T cell therapy as well as the patient’s endogenous immune system represent two untapped pools of anti-tumour bystander cells. Recruiting both of these components could potentially generate a more concerted immune response against the tumour, and most importantly, target antigen-negative tumours that are otherwise inaccessible, thus enhancing the overall efficacy of CAR T cells against solid malignancies. To achieve bystander recruitment, CAR T cells have been further engineered to secrete either bispecific T cell engagers (BiTEs) or immune stimulating molecules, with promising preclinical data reported to date.

#### 2.3.1. Secretion of Immune Cell Engagers

BiTEs are fusion proteins consisting of two scFv fragments; one specific for surface markers ubiquitously expressed on effector cells—CD3 for example—and the other specific for antigens highly expressed on the tumour, such as mesothelin, the disialoganglioside GD2, or EGFR. These molecules form an artificial immunological synapse between the effector and target cell that circumvents TCR specificity and results in a targeted release of the cytolytic effector molecules granzyme B and perforin.

Although IV infusion of BiTEs as a stand-alone therapy has shown some degree of clinical benefit against certain cancers, its efficacy is limited by their inability to actively home to a tumour, a significantly short half-life, and an incapacity to self-amplify [87]. Engineering immune cells to constitutively secrete BiTEs, however, would result in targeted delivery and continuous production that is dependent on the tumour homing and longevity of the T cells. More importantly, unlike CAR T cell-based therapies in which only the transduced CAR-expressing cells are tumour-targeting, all cells within a BiTE-secreting T cell product—both non-transduced and transduced—as well as the patient’s own cytotoxic immune cells, will have a redirected tumour-targeting potential. In addition, because BiTEs lack endodomains, they potentially do not induce the tonic signalling that is often associated with CAR expression [88], which may alleviate exhaustion and prolong survival. It is important to note, however, that BiTEs only stimulate effector cells through CD3 cross-linking, and thus lack the potent costimulatory signals from CAR endodomains, potentially resulting in a shorter-lived response.

Preclinical studies have assessed the feasibility of engineering T cells with BiTE-secreting capabilities, either alone or in combination with a CAR. Iwahori et al., for example, engineered T cells to secrete BiTEs specific for CD3 and the tumour-associated antigen EphA2, a receptor tyrosine kinase overexpressed in a broad range of solid cancers that has recently been implicated in tumour metastases. BiTE-secreting cells were cytotoxic against EphA2-positive tumour cell lines in vitro and capable of recruiting cytotoxic bystander T cells. Similarly, in severe combined immunodeficiency disease (SCID) mice bearing EphA2-positive A549 lung tumours, expansion of both transduced and non-transduced cells was observed, further supporting a bystander recruitment effect. The amount of BiTE produced by the T cells was strongly linked to how activated they were, i.e., more antigenic stimulation of the engineered T cells resulted in higher levels of BiTE production and thus more bystander T cell activation. In terms of in vivo efficacy, a reduced tumour burden was observed in two immunodeficient xenograft models: intracranial EphA2-positive U373 cells and IV EphA2-positive A549 cells. T cells secreting EphA2- or CD19-targeting BiTEs were tested and efficacy was observed with the EphA2 construct only.

In contrast, Choi et al. used a dual engineering approach to co-express an EGFRvIII-targeting CAR in conjunction with an EGFR/CD3-targeting secretable BiTE [89]. EGFRvIII is a glioblastoma-specific tumour antigen found only on malignant cells; however, its expression is heterogenic, and therefore CAR T cells are unable to eradicate all of the tumour, resulting in outgrowth of the EGFRvIII-negative cells. Combined with a BiTE specific for the wild-type EGFR, which is frequently overexpressed in glioblastoma, BiTE co-expression has the potential to recruit bystander cells and redirect them against wild type EGFR, thus resulting in complete tumour control. This hypothesis was elegantly tested in an orthotopic xenograft model consisting of NSG mice challenged with an intracranial injection of U251. In this model, T cells expressing CAR only were unable to kill all of the tumour, which resulted in an outgrowth of EGFRvIII-negative cells, whereas BiTE co-expression resulted in complete tumour control.

#### 2.3.2. Secretion of Immune Orchestrating Cytokines

Engineering immune cells with the ability to secrete particular cytokines is an alternative approach to enhancing bystander activation. Gene cassettes that encode cytokines typically contain fewer base pairs than those encoding BiTEs, potentially making them more compatible with polycistronic gene delivery systems and also easier for immune cells to transcribe.

Recent efforts have focussed on IL-12 and IL-18, which are both pleiotropic myeloid-derived cytokines capable of priming naïve CD8^+^ T cells and stimulating innate-like cells such as NK cells, NKT cells, and γδ T cells [90,91,92]. They can also activate myeloid cells; for example, IL-12 and IL-18 enhance dendritic cell (DC) antigen presentation and also re-polarise TAMs towards an anti-tumour M1 pro-inflammatory phenotype [93,94]. The secretion of IL-12 or IL-18 by CAR T cells may be able to raise local concentrations of these cytokines within the TME, potentially enhancing CAR T cell function itself, as well as creating a more favourable immune landscape within the tumour.

The feasibility of engineering IL-18-secreting CAR T cells has already been tested [95,96,97]. Avanzi et al. treated an immunocompetent syngeneic mouse model with murine CD19-targeting CAR T cells constitutively secreting IL-18 and observed enhanced persistence, tumour infiltration, and overall survival compared with CAR only controls [96]. Importantly, only dual engineered T cells were able to modulate the TME, as shown by expansion of endogenous effector cell populations—namely NK cells, NKT cells and CD8^+^ T cells—as well as maturation of DCs and M1 repolarisation of TAMs. Consistent with this bystander activation, the therapy was able to reject both CD19-positive and CD19-negative tumours.

Chmielewski et al. generated carcinoembryonic antigen (CEA)-targeting CAR T cells with an inducible IL-12 construct [98]. To assess the efficacy of these cells, the authors used a lymphocyte-deficient mouse model that has a fully functioning macrophage compartment. In this model, the IL-12-secreting CAR T cells prevented tumour formation by CEA-positive tumours, and more importantly, prevented formation of CEA-negative tumours in the presence of CAR signalling. Interestingly, rejection of CEA-negative tumours was macrophage and TNF-α dependent, suggesting that IL-12 release inside the tumour recruits and stimulates macrophages with an anti-tumour phenotype.

In addition to IL-12 and IL-18, CAR T cells engineered to secrete IL-7, IL-15, and IL-21 (covered in more detail in the next section), although primarily intended to enhance the persistence and activity of the CAR T cells themselves, will also stimulate anti-tumour responses in bystander cells. Of particular note is a study by Lanitis et al., which tested murine IL-15-secreting CAR T cells in a fully immunocompetent syngeneic mouse model, thus allowing for the activation of bystander immune cells to be properly assessed [99]. This model consisted of C57BL6 mice challenged subcutaneously with the B16 melanoma cell line and then treated with murine T cells engineered to co-express tumour vasculature-targeting CARs and secretable murine IL-15. IL-15-secreting CAR T cells had enhanced effector functions, increased levels of the anti-apoptotic protein Bcl-2, reduced expression of PD-1, and demonstrated greater tumour control and persistence in vivo. More importantly, analysis of the endogenous immune infiltrate of tumours showed increased NK cell activation and reduced numbers of M2-like macrophages, suggesting the IL-15-secreting CAR T cells are able to disrupt and reprogram the TME.

#### 2.3.3. Stimulation of Adaptive Immunity

CAR T cell-mediated apoptosis of malignant cells releases antigenic material for infiltrating antigen presenting cells (APCs) to endocytose, process, and cross-present on human leukocyte antigen (HLA) class I to naïve CD8^+^ T cells that circulate through the draining lymph nodes. This has the potential to generate memory cytotoxic CD8^+^ T cell responses, and thus adaptive immunity against the tumour, which may support CAR T cell efficacy against solid tumours and provide long term protection against relapse.

APCs require maturation signals in order to effectively prime naïve T cells. This causes upregulation of CCR7 for lymph node homing, increased expression of costimulatory molecules for T cell activation, and secretion of IL-12p70 for naïve T cell differentiation. APC maturation is typically induced by pattern recognition receptor (PRR)-mediated recognition of pathogen-associated molecular patterns (PAMPs). Because tumours are in essence part of self, and thus “sterile”, maturation of tumour-infiltrating APCs is often suboptimal, which can result in unsuccessful T cell priming and even antigen tolerance. CAR T cells engineered to express APC-activating proteins—such as CD40 ligand (CD40L), TLR agonists, or cytokines—may provide that extra kick required for full APC maturation within the tumour.

Curran et al. co-expressed CD19-targeting CAR T cells with constitutive CD40L expression [100]. CD40L is a cell surface protein transiently upregulated on activated T cells that interacts with its cognate receptor CD40 on the surface of APCs such as DCs, macrophages, and B cells. Although this engineering strategy was originally intended to boost direct killing of malignant B cells through CD40 ligation, it may also enhance CAR T cell efficacy against solid cancers by promoting maturation of tumour-infiltrating DCs, upregulating CCR7, CD80/CD86, and IL-12p70 required for lymph node homing and naïve T cell priming. Indeed, using syngeneic mouse models of B cell lymphoma, Kuhn et al. showed that CD40L-modified CD19-targeting CAR T cells licensed APCs and enhanced endogenous anti-tumour T cell responses in vivo [101]. However, whether or not the constitutive expression of CD40L on CAR T cells can induce maturation of bystander APCs within solid tumours and subsequently result in activation of adaptive immune responses has yet to be demonstrated.

In a recent publication by Johnson et al., CAR T cells were armoured with the co-expression of an endogenous RNA, designated RN7SL1, which activates the principal cytosolic PRRs for viral dsRNA RIG1 and MDA5 [102]. These CAR T cells not only displayed improved proliferation, reduced exhaustion, and better persistence in in vivo models, they also transferred the viral RNA to bystander immune cells via extracellular vesicles, reducing suppressive myeloid cells and promoting inflammatory DCs within the TME of syngeneic mouse models. The concerted effects on both CAR T cells and bystander myeloid cells resulted in improved efficacy against the tumour, and importantly, enhanced the activity of endogenous CD8^+^ T cells.

In conclusion, preclinical studies have shown that CAR T cells can be engineered with added bystander recruitment by exploiting a variety of different immunological pathways. These CAR T cells function as a double-edged sword, not only capable of direct cytotoxic responses against the tumour, but also orchestrating the anti-tumour responses of bystander cells, both the residual non-transduced component of the CAR T cell product and the recovering immune system of the patient (Figure 1). Tapping into and harnessing these bystander cells may elicit a more concerted anti-tumour immune response for better efficacy against solid malignancies.

### 2.4. Survival and Persistence

Persistence of cell-based therapies in vivo is an important attribute that is essential for achieving optimal efficacy, particularly when targeting solid malignancies whereby the cells must endure the unfavourable and challenging conditions of the TME. To this end, CAR constructs have been supplemented with gene cassettes that encode proteins that confer sustained proliferation and resistance to activation-induced cell death (AICD). Constructs that prolong CAR T cell survival long enough to allow for complete eradication of the tumour as well as prevent outgrowth of any residual disease and/or metastatic lesions could prove highly effective against solid malignancies.

Recent efforts have focussed primarily on co-expressing gene cassettes that encode cytokines, particularly those which signal intracellularly via certain Janus Kinase (JAK) and Signal Transducer and Activator of Transcription (STAT) proteins. Specifically, JAK 1/3 and STAT 1/3/5 upregulate the expression of genes associated with proliferation, survival and apoptosis resistance [103]. Cytokines with receptors which share the common gamma chain CD132, i.e., IL-2, IL-4, IL-7, IL-9, IL-15, and IL-21, are all potent stimulators of these pathways [103,104]. With the exception of IL-4 and IL-9, which are associated with the induction of Th2 immunity [105], these cytokines are promising candidates for CAR T cell armouring.

There are three key approaches to engineering CAR T cells with enhanced cytokine production. Firstly, cells can be engineered to constitutively express the cytokine, which, although has the added benefit of providing cytokine support during transit to the tumour, may result in toxicity against healthy cells and require some form of safety suicide switch such as the Rapamycin-activated caspase 9 gene or the CD34/CD20 epitope RQR8 [106,107]. Secondly, cytokine secretion can be controlled in a way that results in targeted release inside the tumour; for example, cytokine expression can be triggered by a CAR or some other tumour-targeting receptor. Thirdly, instead of expressing the cytokine itself, components of the intracellular signalling domains of cytokine receptors can be incorporated into the endodomains of the CAR.

#### 2.4.1. CARs with Cytokine Receptor Endodomains

IL-2 is a well characterised cytokine that promotes proliferation, survival, and cytotoxicity in effector cells such as NK cells, γδ T cells, and CD8^+^ T cells. Although CAR T cells have been shown to secrete some degree of IL-2 upon target cell recognition, further boosting IL-2 signalling may provide a way of enhancing survival. Tregs, however, are potently stimulated by IL-2 due to constitutive expression of the high affinity IL-2Rα chain CD25, thus engineering CAR T cells to secrete more IL-2 may be counterproductive.

To avoid collateral stimulation of Tregs while still exploiting the pro-survival benefits of IL-2, intracellular signalling components of the IL-2 receptor have been incorporated into the endodomains of CARs. Kagoya et al. designed a new generation of CD19-targeting CARs that encode a truncated cytoplasmic domain from the IL-2R chain and a STAT3-binding tyrosine-X-X-glutamine motif together with the TCR signalling (CD3ζ) and co-stimulatory (CD28) domains [108]. The resulting CAR T cells displayed antigen-dependent JAK/STAT kinase activation, which enhanced proliferation and terminal differentiation in vitro, and improved the persistence and efficacy in xenograft mouse models of both solid and haematological cancers.

#### 2.4.2. Secretion of Cytokines

##### IL-7

IL-7 is a haematopoietic growth factor that is typically produced by stromal cells of the bone marrow and thymus. Through binding to its dimeric receptor, which is composed of the common gamma chain paired with the IL-7Rα chain (CD127), it induces haematopoietic stem cells (HSCs) to differentiate into lymphoid progenitors, and stimulates proliferation in T cells, B cells, and NK cells [109]. Importantly, Tregs typically express relatively low levels of CD127 and thus are less responsive to IL-7 [110]. These features make IL-7 a promising alternative to IL-2 for CAR T cell armouring.

Duan et al. generated B cell maturation antigen (BCMA)-targeting CAR T cells that constitutively secrete IL-7 and CCL19 [111]. These cells displayed better expansion, differentiation, migration, and cytotoxicity compared with CAR only controls and are currently being trialled in the clinic (NCT03778346) for refractory/relapsed multiple myeloma, with safety and efficacy observed in the first two patients enrolled. More importantly, in the context of solid malignancies, IL-7- and CCL19-secreting mesothelin-targeting CAR T cells displayed a low expression of markers associated with T cell exhaustion, including PD-1 and TIGIT, and had enhanced efficacy against orthotopic pre-established malignant mesothelioma xenograft models, as well as patient-derived xenograft models of mesothelin-positive pancreatic cancer [112]. Interestingly, the transfer of IL-7/CCL19-producing CAR T cells resulted in an increase in both CAR T cells and non-CAR T cells within the tumours, suggesting that IL-7 secretion also confers a bystander effect [112].

##### IL-15

IL-15 has captured huge attention as a potential CAR T cell armouring cytokine due to its potent stimulatory effects on NK cell, γδ T cell, and memory CD8^+^ T cell effector responses, such as proliferation, survival, and cytotoxicity, and lack of concomitant Treg stimulation and T cell exhaustion [113].

IL-15 is typically produced by cells of the myeloid lineage, such as monocytes, macrophages, and DCs, and signals through a trimeric receptor composed of CD132, IL-15/IL-2Rβ chain (CD122), and the high affinity IL-15Rα chain (CD215). Signalling is unique in that CD215 is typically expressed on myeloid cells and trans-presents IL-15 to CD122/CD132 dimers on NK cells, γδ T cells, and CD8^+^ T cells.

Promising efficacy against solid tumours has been observed in preclinical and clinical studies using systemic administration of IL-15 or IL-15 complexes such as ALT-803, a complex composed of a human IL-15 superagonist variant and a human IL-15Rα sushi domain-Fc fusion protein [114,115,116]. Severe toxicities, however, were recorded, demonstrating the need for stringent dose escalation or even more targeted delivery. Engineering CAR T cells to secrete their own source of IL-15—whether by constitutive or inducible gene expression cassettes—may greatly enhance efficacy in vivo without the accompanying toxicity associated with systemic delivery.

Preclinical studies have demonstrated the proof-of-concept for IL-15-armoured CAR T cells using haematological and solid tumour models. Hurton et al., for example, generated CD19-targeting CAR T cells that co-expressed a membrane bound construct of IL-15 composed of the IL-15 protein fused to IL-15Rα [117]. These CAR T cells displayed constitutive STAT signalling that was independent of CAR stimulation, resulting in enhanced long-term persistence both in vitro and in vivo. Importantly, these cells were better equipped to reject NALM-6 engraftment in NSG mice over and above the CAR only controls, and persisted long after the tumour was successfully cleared. Similarly, Miller et al. showed enhanced efficacy using CD19-targeting CAR T cells armoured with an inducible single chain IL-15 superagonist—composed of IL-15 tethered to the sushi domain of the IL-15Rα subunit—under the control of the temperature-sensitive transcription factor Heat Shock Factor 1 (HSF1) [118].

Liu et al. engineered cord blood-derived NK cells to co-express CD19-targeting CARs, secretable IL-15, and a suicide safety switch [119]. These cells markedly improved survival in a RAJI xenograft mouse model [119] and are undergoing an early phase clinical trial (NCT03056339) consisting of 11 patients with relapsed or refractory CD19^+^ cancers (lymphoma and CLL) [120]. Expansion and persistence have been observed, and clinical responses obtained in the majority of patients.

Of particular interest is a preclinical study by Chen et al., which tested the concept of IL-15-armoured CAR T cells against solid tumours. Using neuroblastoma in vitro and in vivo models and GD2-targeting CAR T cells with or without secretable IL-15, the authors demonstrated that IL-15 co-expression resulted in reduced exhaustion, an enriched stem cell-like phenotype, improved survival, and a repeated anti-tumour effect upon rechallenge [11].

##### IL-21

IL-21 is a pleiotropic cytokine that signals through dimeric receptors composed of the common gamma chain and the IL-21Rα chain expressed on both myeloid and lymphoid cell lineages. IL-21 primarily promotes anti-tumour responses by stimulating proliferation, survival, differentiation, and cytotoxicity in CD8^+^ T cells and NK cells [121,122]. Despite also inducing retinoic acid receptor-related orphan receptor-γt (RORγt) [123], a transcription factor that regulates the pro-tumour Th17 phenotype in CD4^+^ T cells, IV bolus injections of IL-21 have shown favourable outcomes against solid tumours in early phase clinical trials [124,125], indicating the potential benefit of this cytokine as an anti-tumour agent.

Stach et al. co-expressed IL-21 in CD19- and PSA-targeting CAR T cells and observed a more stem-like phenotype with increased antigen-specific expansion and reduced apoptosis in vitro [126]. In vivo, these cells displayed enhanced tumour infiltration and reduced tumour growth. Although the effects of IL-21 are likely to be multifaceted, the authors report that IL-21 was able to directly interfere with the immunosuppressive effect of CLL cells towards CAR T cells, thus explaining—at least in part—how this cytokine may boost CAR T cell function.

#### 2.4.3. Expression of Anti-Exhaustion Intracellular Signalling Molecules

An alternative strategy is to constitutively express genes that encode pro-survival intracellular signalling proteins. Akt/protein kinase B, for example, is a serine/threonine-specific protein kinase that promotes T cell proliferation and survival via multiple anti-apoptotic signalling pathways. Sun et al. co-expressed Akt/protein kinase B in GD2-targeting CAR T cells and showed increased levels of the anti-apoptotic molecules Bcl-2, Bcl-xL, and Mcl-1 [127]. These cells also produced higher levels of IFN-γ and IL-2, displayed elevated NFκB activity, resisted apoptosis following tumour target exposure, and were more potent at killing LAN-1 cells in vitro. Similarly, Lynn et al. engineered CAR T cells to overexpress c-jun protein, a transcription factor that forms a number of dimeric complexes, collectively termed activator protein-1, which positively regulates cell proliferation [128]. The authors demonstrated that c-jun overexpression in CAR T cells enhanced their proliferative capacity as well as their potency in multiple xenograft mouse models of haematological and solid tumours.

Collectively, these studies demonstrate that the co-expression of genes associated with persistence confer a functional benefit to CAR T cells as well as bystander cells (Figure 1). This approach has great potential in prolonging the survival of CAR T cells in vivo, and thus improving the overall efficacy against solid tumours.

## 3. Engineering Techniques for Delivering Polycistronic Gene Constructs

Engineered immune cells have the potential to overcome the challenges associated with solid tumour targeting. Preclinical studies have tested the feasibility of co-expressing CARs with an additional armouring transgene conferring either of the following: (a) enhanced homing, (b) improved counter defence mechanisms against immunosuppression, (c) augmented capacity to activate bystander cells, or (d) prolonged survival in the adverse conditions of the TME. It is likely, however, that more than one armouring transgene is needed to generate CAR T cells with all the necessary adaptations required for eradicating solid tumours. Genetic engineering strategies for delivering large constructs that contain multiple transgenes are, therefore, hugely important for creating the next generation of cell-based therapies (Figure 2).

Different gene delivery tools can be used to engineer immune cells for adoptive cell therapy, including non-viral techniques using either electroporation, CRISPR-Cas9, or transposons (e.g., Sleeping Beauty and piggyBac), as well as viral-based methods utilising adenoviruses, adeno-associated viruses (AAV), or retroviruses (e.g., HIV lentivirus and MLV gammaretrovirus) [129]. These approaches are either (a) non-integrating, i.e., the transgene is not integrated into the host genome and therefore not passed onto daughter cells, or (b) integrating, where the DNA is incorporated into the host genome and subsequently passed onto daughter cells.

Integrating viral-based gene delivery tools composed of either gammaretrovirus or lentivirus backbones are typically used to generate CAR T cells [130]. Although gammaretroviruses are stable and yield high viral titres, they can only carry relatively small gene constructs. Lentiviruses, in contrast, can accommodate larger gene constructs but at the cost of poor stability and lower viral titres, often requiring a concentration step using ultracentrifugation or a viral capture reagent [131]. The capacity for lentiviral vectors to carry larger cargo makes them the ideal candidate for polycistronic gene constructs and thus an important tool in generating the next generation of CAR T cells. In this section, we discuss some of the key methods of engineering cells to express multiple transgenes using a single lentiviral vector.

### 3.1. Multiple Promoters

Potentially the most rudimentary approach to expressing multiple genes in a single cell is to use multiple promoters, i.e., transduce a cell with a single gene construct that contains a separate transcription unit for each transgene it contains. This results in the synthesis of each gene being completely separate, thus avoiding the fusion of gene products that can often occur with other strategies. Using multiple promoters, however, results in a very large construct that is typically not practical for polycistronic platforms consisting of multiple genes of interest. A more favourable approach is to use a single vector containing multiple genes under the control of a single promoter. There are three key methods of separating the gene products either pre or post synthesis: (1) proteolytic cleavage, (2) internal ribosome entry sites (IRESs), and (3) self-cleavage peptides.

### 3.2. Single Promoter

#### 3.2.1. Proteolytic Cleavage

Proteolytic cleavage involves separating multiple genes within a single-vector polycistronic construct with short linker sequences that encode cleavage sites for enzymes ubiquitously expressed in the host cell. The protein products are translated together and synthesised as one multimeric fusion protein that is subsequently cleaved into its individual proteins by proteolytic enzymes specific for the integrated cleavage sites. Furin, for example, is a constitutively expressed golgi endoprotease that has been used to successfully express multiple cDNA sequences. Gaken et al., for instance, transduced target cells with a single gene construct containing cDNA for IL-12p40, IL-12p35, and IL-2, separated by two furin cleavage sites [132]. The relatively small length (12 base pairs) of a furin cleavage site makes it a useful approach for building polycistronic gene constructs. Furthermore, co-expression is guaranteed and the post-synthesis cleavage results in a clean break with no extra superfluous amino acids left on the proteins of interest. Proteolytic cleavage is, however, dependent on the cellular expression and function of the cleavage enzyme.

#### 3.2.2. Internal Ribosomal Entry Sites

IRESs are viral-derived DNA sequences that have been used successfully to separate different genes within polycistronic constructs [133]. The sequences cause dissociation of ribosomal complexes from the mRNA strand and subsequent formation of new translation initiation complexes that continue translation [134]. Consequently, translation of genes upstream and downstream of an IRES sequence occur independently. In a tri-cistronic construct containing three genes separated by two IRES sequences, for example, translation of the first gene will be IRES-independent, whereas translation of the second and third gene will be IRES-dependent. The expression of genes that are translated in an IRES-dependent manner (i.e., genes two and three in the hypothetical tricistronic construct) is equal, regardless of the sequence order; however, IRES-dependent translation is often markedly reduced compared with IRES-independent, which results in a higher expression of the first gene and a reduced expression of the second and third genes. Furthermore, IRES sequences are relatively large (typically >500 base pairs) and thus cumbersome to incorporate into polycistronic constructs. Nonetheless, IRES is a useful tool for expressing multiple genes in a single vector, consistently resulting in complete separation of the gene products with no excess material added to the proteins.

#### 3.2.3. Cleavage Peptides

The so-called 2A self-cleaving peptides are a group of viral-derived oligopeptides composed of short amino acid sequences 18-22 base pairs long that mediate the separation of polypeptides during translation. Four 2A cleavage peptides have been identified and used successfully in bi-, tri-, and quadcistronic gene constructs [135]. F2A is from the foot and mouth disease virus, E2A is from the equine rhinitis A virus, P2A is from the porcine teschovirus-1 virus, and T2A is from Thosea asigna virus. Although these sequences are called cleavage peptides, they do not involve cleavage per se. Instead, they create a ribosomal skip by forming a glycyl-prolyl peptide bond at the C-terminus of the 2A. After the synthesis of the first gene is complete, synthesis continues without the dissociation of ribosome, resulting in continuous synthesis of two proteins.

Cleavage peptides are relatively small in size, they have good cleavage efficiency, and importantly, they result in consistently high levels of downstream protein expression. However, inhibition of 2A can sometimes occur, thus resulting in the production of uncleaved fusion proteins [136]. Furthermore, 2A cleavage peptides leave an additional 18–21 amino acids on the C-terminus of the preceding protein and an extra proline on the N-terminus of the subsequent protein, which may interfere with protein function. Due to the relatively small size of proteolytic cleavage sites, however, sequences encoding furin recognition sites can be easily inserted upstream of 2A in order to remove these redundant 2A residues [137,138].

## 4. Conclusions

Over the past decade, we have seen huge advances in the use of cell-based therapies for the treatment of cancer. Whereas previously cell-based therapies were limited to using non-engineered cells, which would struggle to survive beyond in vitro expansion and rapidly succumb to the immunosuppressive effects of the tumour in vivo, we now have the technology to engineer immune cells with selected genes that markedly bolster their effector function in vivo and counteract the adverse conditions of the TME. Moreover, we can create polycistronic constructs capable of stably transducing a cell with multiple genes simultaneously, providing a continuous supply of natural and/or recombinant gene products for optimal performance.

In addition, approaches to engineering immune cells for adoptive cell therapy are advancing beyond just membrane-bound cytotoxicity receptors that only enhance the tumour targeting capacity of the cells that express it, and more towards secretable immunotherapy payloads, including effector cell engagers, checkpoint blocking antibodies, survival promoting cytokines, and immune orchestrating cytokines. This new wave of adoptive cell therapy uses immune cells as tumour homing cargo vessels for the targeted delivery of both membrane-bound and secretable immunotherapeutic agents directly into the TME. Importantly, these cells are no longer confined by direct cell–cell contact and can elicit their effects at range, disrupting the immunosuppressive cocoon of the tumour and converting it into a pro-immunogenic environment for anti-tumour immune cells to thrive, thus generating an orchestrated immune response against the tumour.

Although the majority of research in this field has focussed primarily on armouring αβ T cells, many of the engineering concepts described in this review are applicable to other immune chassis. Non-alloreactive cytotoxic effector cells such as NK cells and γδ T cells are currently being developed as a platform for “off-the-shelf” engineered cell-based products [139,140,141,142,143]. Furthermore, the potential to express CARs in macrophages and B cells has also been tested, exploiting alternative immune cells that function in antigen presentation, phagocytosis, and antibody production [144,145,146].

Through the co-expression of certain genes in the appropriate immune cell chassis, there is potential to disrupt the TME and improve the overall efficacy of cell-based therapies against solid malignancies. Although huge advances have been made, we have yet to see whether these approaches are effective in the clinic and many substantial hurdles still remain; in particular, current state-of-the-art engineering vectors and GMP-compliant large-scale manufacture of transduced cells for adoptive transfer present ongoing technical and financial challenges. More importantly, the rare but life-threatening toxicities associated with side effects such as cytokine release syndrome and on-target off-tumour cytotoxicity pose an underlying safety concern. Furthermore, technologies such as CARs and BiTEs are highly dependent on tumour antigen expression and thus can be blunted by the phenomena of antigen loss often associated with solid tumours. Downregulation of MHC class I expression and outgrowth of antigen low/negative malignant cells following immune selection pressure remain important limitations in this field and areas of ongoing research.

## Figures and Tables

**Figure 1 cancers-13-06000-f001:**
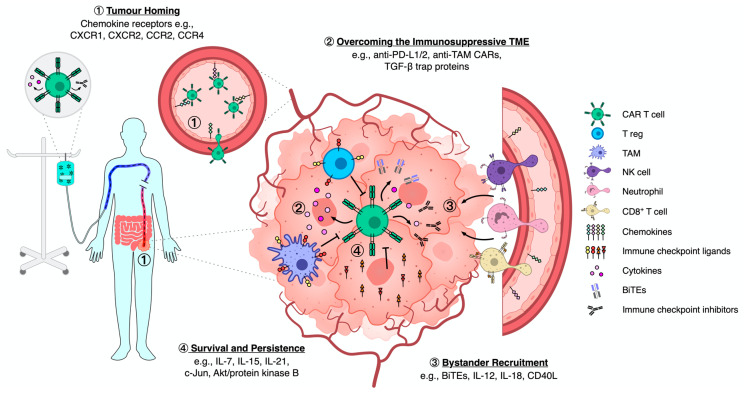
Schematic diagram outlining CAR T cells homing to and interacting within a solid tumours’ immunosuppressive TME. Examples of engineering strategies are listed within each numbered section. ① Following IV infusion, engineered CAR T cells are dispersed throughout the total blood compartment of the patient with only a minority reaching the tumour vasculature for potential tissue homing. Upon entry into the tumour vasculature, CAR T cells are exposed to a plethora of soluble and endothelial-bound chemotactic molecules that mediate endothelial tethering, extravasation and interstitial migration into the TME. ② Upon successful entry into the TME, CAR T cells face immunosuppressive barriers that circumvent anti-tumour efficacy. Cancer cells, as well as immune populations including TAMs and Tregs, express inhibitory molecules and secrete anti-inflammatory factors that promote immune evasion and tumour progression. Although surrounded by immunosuppressive cues, CAR T cells will constitutively produce and secrete the proteins they are engineered to do so; strategies used to counteract the immunosuppressive TME or promote anti-tumour efficacy include pro-inflammatory cytokines, BiTEs and immune checkpoint inhibitors. ③ Following sustained transduced protein production, particularly in the context of pro-inflammatory cytokines, bystander effector immune cells are recruited from both the peripheral blood and neighbouring stromal compartments towards the tumour. Upon performing anti-tumour immune mechanisms, the bystanders will produce further cytokines that self-perpetuates recruitment in a positive feedback loop. Aside from further recruitment, the ongoing cytokine production from engineered CAR T cells and engaged bystanders will promote repolarisation of the TME away from an immunosuppressive state. ④ CAR T cells can also be engineered to produce cytokines and intracellular signalling proteins that promote survival, persistence and, therefore, anti-tumour efficacy within the TME.

**Figure 2 cancers-13-06000-f002:**
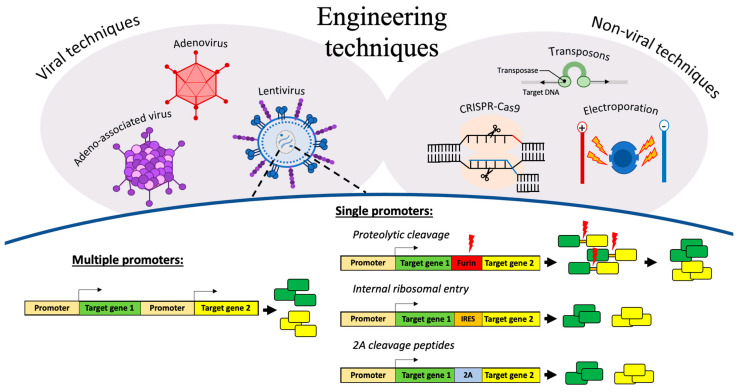
Gene delivery tools for immune cell engineering including delivering polycistronic gene constructs in lentiviral vectors. Different gene delivery tools include non-viral techniques and viral-based methods. As lentivirus can carry larger cargo, it is possible to engineer cells to express multiple transgenes. This can be done either using multiple promoters, which can result in a very large constructs, or single promoters where the gene products can be separated with proteolytic cleavage sites, internal ribosomal entry sites, or 2A cleavage peptides.

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
