# Peer review of "Payload Delivery: Engineering Immune Cells to Disrupt the Tumour Microenvironment"

_cancers, 2021, doi:10.3390/cancers13236000_

Round 1

Reviewer 1 Report

I am attaching a PDF document with a few comments.

Author Response

Dear Reviewer 1,

Thank you for taking the time to review our manuscript and for providing helpful comments - I have addressed each of them in turn below:

  • We have mentioned combination therapy with anti-CTLA-4 in section 2.2.3 on page 9 of the revised manuscript as suggested.
  • Line 2 of the simple summary on page 1 of the revised manuscript has been amended from “receptors have proven” to “receptors have been proven”.
  • “Hampers” on page 8 of the revised manuscript has been changed to “dampen”. Also, we noticed the word hamper was used multiple times throughout the review and have used synonyms to reduce the use of the word. Two uses of “hamper” on page 7 have been changed to “impede” and “suppress”.
  • Font colour in the revised manuscript has been corrected. This change has not been tracked.

I hope that you find these changes satisfactory, and look forward to hearing your response.

Kind regards

Dr Jonathan Fisher

Reviewer 2 Report

This is a well written review on how to improve CAR T cell efficacy in solid tumors. It is timely and comprehensive.  The claims are supported by the references aside from the half a dozen specific comments below. The tone is balanced, but the review could be improved by the following comments. I would be supportive of publication after modification.

Major concerns:

  1. Please consider placing their focus within the larger context of “armoring” other immune cell types including NK cells (only 2 NK references included) or APC cells.
  2. Please consider a more balanced discussion by mentioning the rare but life-threatening toxicities associated with this class of therapeutics.
  3. MHC downregulation or loss is observed in solid malignancies. Antigenic drift is observed under immune pressure. Please incorporate a discussion of limitations of CAR T cells in these settings.
  4. “Indeed, clinical trials have tested the efficacy of various autologous and allogeneic non-engineered cell-based products and have yet to demonstrate any meaningful clinical responses.” This sentence ignores the entire field of graft vs disease which is well recognized therapeutic concept in clinical medicine.
  5. “Vascularisation of a solid tumour provides the all-important access required for an intravenously administered cell-based therapy, provided it is programmed with the correct tissue homing required to exit the circulation at the precise anatomical location of the tumour.” This ignores the concept, that has at least been partially experimentally borne out, by Dr. Duda and others that vascular normalization actually promotes T cell trafficking. Suggest revision of concept and inclusion of additional references.
  6. “Moreover, their cognate ligands including CCL2, CCL3, CCL4, CCL5, CXCL9, CXCL10 and CXCL11, are often upregulated in a range of different tumour types24-28.” While this is true, the converse is also true- Dr. Weiping Zou and others have shown chemokine silencing is a hallmark of immune invasion. Recommend qualifying assertion and adding additional references.

Minor concerns

  1. Reference one speaks only of NK cells, while the scope of the sentence it is used in is more broad. Recommend adding additional references specifically speaking to CD8 T cells.
  2. “Genes encoding chemokines that facilitate tumour homing4,” That concept is not supported by the provided reference.
  3. “CD25-bearing jurkats” this phrasing is a too casual.
  4. References 60-63 could be improved.
  5. Reference 64 should be replaced by more relevant work, ex. https://pubmed.ncbi.nlm.nih.gov/29337305/ or others.
  6. “reduced expression of the exhaustion marker PD-1”. PD-1 is expressed by activated T cells as well as exhausted T cells. It is not an exhaustion marker.

Figure 1. Consider adding an aorta.

Figure 2 mentions Crispr. There is no mention of this approach in the text. It should be added.

This is a small note, and no change is required, but the title is a mixed metaphor.  

Author Response

Dear Reviewer 2

Thank you for taking the time to review our manuscript and for providing helpful comments and suggestions - they really added to the overall discussion.  Below is a list of our responses, separated into the major and minor points which you raised:

Major concerns

  1. We have added a paragraph to the discussion on page 22 of the revised manuscript to broaden the discussion to include engineering of non-alpha beta T cells.
  2. We have added a statement to the discussion on page 22 of the revised manuscript mentioning the toxicities associated with this therapeutic approach.
  3. As suggested by the reviewer we have added discussion of antigen loss in the last paragraph of the discussion section.
  4. We have removed “autologous and allogeneic” from the second paragraph of the introduction.
  5. We have added the concept of normalisation of tumour vasculature and a relevant reference to the last paragraph of section 2.1 in the revised manuscript.
  6. Epigenetic silencing of chemokines as a tumour escape mechanism has been added to the last paragraph of section 2.1 in the revised manuscript.

Minor concerns

  1. An additional reference has been added to support the second sentence in the introduction.
  2. Additional references have been added to support the sentence “Genes encoding chemokines that facilitate tumour homing” on page 5 of the revised manuscript.
  3. “CD25-bearing” has been changed to “CD25-expressing” on page 9 of the revised manuscript.
  4. Additional references have been added to the first sentence of the second paragraph of section 2.2.3 on page 9 of the revised manuscript as suggested by the reviewer.
  5. The reference supporting the statement “Tumour cells and tumour-associated myeloid cells such as TAMs and MDSCs can express high levels of checkpoint ligands such as PD-L1 and PD-L2” on page 9 of the revised manuscript has been amended as suggested.
  6. Use of the term “exhaustion markers” has been changed to “markers associated with T cell exhaustion” on pages 10 and 15 of the revised manuscript. The word “exhaustion” was also deleted from page 13 of the revised manuscript.

Figure 1: We have changed the colouring and added a line break to the vasculature in the drawing of the patient in figure 1 to depict transition from venous to arterial blood  

Figure 2: “CRISPR-Cas9” has been added to the text on page 19 in the revised manuscript. “Crispr-cas9” in the figure has also been changed to “CRISPR-Cas9”.

Title: We have changed “armoured” to “engineered” in the title.

We hope that these revisions meet with your approval, and look forward to hearing your responses.

Kind regards

Dr Jonathan Fisher

Reviewer 3 Report

This review by Daniel Fowler et al. provides an overview of the recent progress in CAR-T cells for treating cancers. It focuses on two parts of this field, one is how the novel ways were used to engineer the immune cells to enhance efficacy against solid tumors and the other is the novel gene engineering techniques for delivering immunotherapeutic payloads into the tumor microenvironment. This is a very interesting topic and this article gives a scientific perspective on this topic. This is a well-organized and well-written article with appropriate literature citations.

Minor points to consider before publication:

  1. Page 16, it should be “3.2.3”, not “3.2.3.2”.
  2. The abbreviation of hydrogen peroxide should be “H2O2”, not “H202”.

Author Response

Dear Reviewer 3,

Thank you for taking the time to review our manuscript.  We have made the following changes in response to your comments:

  • We have removed 2A from the section title on page 20 of the revised manuscript to avoid confusion.
  • H202 changed to H2O2 throughout the document.

We hope that these changes meet with your approval, and we look forward to hearing your response.

Kind regards

Dr Jonathan Fisher